# Research Progress in J-Proteins in the Chloroplast

**DOI:** 10.3390/genes13081469

**Published:** 2022-08-17

**Authors:** Lu Zhao, Ting Jia, Qingsong Jiao, Xueyun Hu

**Affiliations:** 1College of Bioscience and Biotechnology, Yangzhou University, Yangzhou 225009, China; 2Joint International Research Laboratory of Agriculture and Agri-Product Safety of the Ministry of Education of China, Yangzhou University, Yangzhou 225009, China

**Keywords:** chloroplast, HSP70, J-proteins, molecular chaperone

## Abstract

The J-proteins, also called DNAJ-proteins or heat shock protein 40 (HSP40), are one of the famous molecular chaperones. J-proteins, HSP70s and other chaperones work together as constitute ubiquitous types of molecular chaperone complex, which function in a wide variety of physiological processes. J-proteins are widely distributed in major cellular compartments. In the chloroplast of higher plants, around 18 J-proteins and multiple J-like proteins are present; however, the functions of most of them remain unclear. During the last few years, important progress has been made in the research on their roles in plants. There is increasing evidence that the chloroplast J-proteins play essential roles in chloroplast development, photosynthesis, seed germination and stress response. Here, we summarize recent research advances on the roles of J-proteins in the chloroplast, and discuss the open questions that remain in this field.

## 1. Introduction

As the classical sessile organisms, plants are exposed to a variety of environmental pressures such as abnormal temperature changes, drought, salt and alkali stress or pathogen infection. To cope with the fluctuating environmental stress conditions, plants gradually acquired systematic protection mechanisms to maintain normal life activities during the long-term evolutionary process. The heat shock protein (HSP) family, which includes HSP100; HSP70(DnaK); HSP90; HSP60; DNAJ proteins (also called J-protein or HSP40); and small HSP, is involved in the process of plants responding to abiotic stress [1,2]. Heat shock proteins are one of the most representative regulatory factors and also a kind of widely existing molecular chaperone [3], which acts at the frontline of defense against protein aggregation and plays an important role in helping plants cope with stressful environments [1].

Chloroplasts are not only organelles for plant photosynthesis, but also act as a general sensor for plants to perceive changes in the cellular or external environment [4]. When plants are subjected to adverse environmental effects such as high-temperature stress, the reactive oxygen species (ROS) accumulated inside of chloroplasts and proteins in chloroplasts are damaged or misfolded, which heavily affects the function of chloroplasts. In order to maintain the normal physiological functions of chloroplasts, plants need to rely on the chloroplast protein quality control system (cpPQC), such as the chloroplast heat shock proteins (cpHSP) [5], to degrade or reactivate the damaged or misfolded proteins. cpHSP70-1 is a major ATP-dependent chaperone that maintains proteostasis in chloroplasts, together with its co-chaperones: one is the co-chaperone chloroplast GrpE (CGE) [6]; the second is presumed to be the co-chaperone J-proteins similar to that in the cytoplasm [7]. Studies have shown that J-protein can improve the binding ability of HSP70 to substrate proteins [1].

All J-proteins contain the J-domain, which is a structure consisting of approximately 70 amino acids with an invariant histidine-proline-aspartic acid (HPD) tripeptide motif. According to their structural domain, J-proteins can be broadly divided into three categories [8,9]. All three classical types of J-proteins (class A, class B and class C types) contain J-domains for interaction with Hsp70 [10]. Some proteins only have a J-domain-like structure without the HPD tripeptide motif. These proteins are called J-like proteins [8]. It is reported that *Arabidopsis* contains two kinds of chloroplast heat shock protein 70 (cpHSP70), while this is true of at least 18 J-proteins and many J-like proteins in chloroplasts [8,11,12]. Except for the function as the co-chaperone of cpHSP70, some chloroplast J-proteins and J-like proteins have been demonstrated to play roles in many different biological processes [8,13,14].

In this review, we briefly introduce recent research progress on the roles of chloroplast J-proteins. The bottleneck and future research directions in the study of chloroplast J-proteins are also discussed.

## 2. DJA5 and DJA6

Both DJA5 and DJA6 have four domains: a J-domain; a glycine/phenylalanine-rich domain; a zinc-finger domain (also known as a cysteine-rich domain, CR domain); and a C-terminal domain [15,16]. They belong to the class A type of J protein. Both of them are localized in the chloroplast [8], and there are only four class A type of J-protein present in the plastid of *Arabidopsis*. The other two are DJA4 and DJA7. When DJA5 or DJA6 was knocked out, the single mutant was compensated by the other protein since the functions of DJA5 and DJA6 overlapped [14]. Therefore, the phenotype and the chloroplast ultrastructure of single mutants did not show obvious differences compared with that of WT. The *dja5dja6* double knock-out mutant is seedling-lethal on soil; it presents an albino phenotype due to reduced chlorophyll under heterotrophic conditions. The leaves of *dja5dja6* were gapped and the seedling developed tiny yellow pedicels, with only a few sterile flowers growing. On the ultrastructure, *dja5dja6* possesses smaller and irregularly shaped chloroplasts and the thylakoid membrane is missing. The authors found that *DJA5* and *DJA6* are co-expressed with chloroplast sulfur utilization factor (SUF) system and the accumulation of chloroplast iron-sulfur (Fe-S) proteins is heavily affected in the *dja5dja6* mutant. the protein content in the photosystem electron transport chain and the chloroplast SUF apparatus also decreased to a certain extent, which hindered the photosystem electron transport process. It can be seen that DJA5 and DJA6 are critical for maintaining a normal plastid shape, the normal phenotype of plant growth and Fe-S protein content [14].

Since excess iron and sulfur are harmful to cells, the biogenesis of Fe-S clusters in the chloroplast needs to be tightly controlled through a complex system [17,18]. During evolution, the SUF assembly system in the plastid was preserved [19]. In SUF systems, the mechanism of sulfur movement and cluster formation during the biogenesis of Fe-S clusters has been intensively studied [18,20,21], while the mechanism of iron utilization in this process is unclear. Further studies found that among the domains of the DJA5 or DJA6 protein is the cysteine-rich domain whose cysteine residues can bind iron. If the cysteine in CR domain was replaced with other amino acids, the binding affinity of DJA5 and DJA6 to iron would be affected, and the Fe-S cluster assembly in the plastid would be affected, accordingly. After binding iron, DJA5 and DJA6 interact with SUF system components such as SUFE1 and SUFC in the chloroplast through their J-domains, and transfer the iron-bound by cysteine residues to the iron receptor SUFD to promote the assembly of Fe-S clusters [14]. Therefore, DJA5 and DJA6 were demonstrated to be the iron donor of the scaffold of Fe-S cluster biogenesis in the plastid.

## 3. DJC75

DJC75 (a J-protein, also named CRRJ and NdhT) and DNAJD15 (a J-like protein, also named CRRL and NdhU) are the subunits of the chloroplast NADH dehydrogenase-like (NDH) complex [22,23,24], which are required for the activity of NDH, functioning in cyclic electron transport [25]. In addition, DJC75 is essential for the accumulation of DNAJD15 and CRR31, an NDH activity-required protein.

Recently, it was found that DJC75 and DNAJD15 cooperate with each other and participate in nitrate-promoted seed germination in the dark [26,27]. This also suggests that the two classes of proteins can act synergistically to regulate life activities. *djc75*, *dnajd15*, *cpHSP70-1* or *cpHSP70-2* mutation heavily abolished the nitrate-stimulated germination of *phyB* mutant seeds, whose germination rate increased significantly under the culture condition containing both micronutrients and nitrate [28,29]. Therefore, it is suggested that DJC75 may activate cpHSP70s ATPase through its HPD (histidine-proline-aspartic acid) motif, so as to recruit cpHSP70s to participate in nitrate-stimulated seed germination in the dark [27]. So far, it is unclear whether NDH is involved in nitrate-stimulated seed germination in the dark. If not, DJC75 and DNAJD15 may play other functions that are independent of NDH. Furthermore, it was found that DJC75 regulates the transcription of *GA30ox1*, a key GA biosynthetic gene, that may contribute to the regulation of nitrate-promoted seed germination in the dark, although the mechanism needs further investigation [27].

## 4. AtJ20, AtJ8 and AtJ11

AtJ20 (DJC26, At4g13830) is a plastid J-protein containing J-domain only. J20 is able to interact with inactive and aggregated desoxyribose5-phosphate synthase (DXS), the first enzyme of the plastidic isoprenoid pathway [5]. It acts as an adaptor that provides its substrate, damaged DXS to cpHSP70s. Thereafter, cpHSP70 can deliver the irreversible inactive DXS to the Clp protease for degradation. Indeed, *AtJ20* knock-out mutant accumulated high levels of DXS protein with reduced levels of DXS enzyme activity, while the transcription of *DXS* was not changed. *j20-1*, *cphsp70-1* and *cphsp70-2* mutants possess higher sensitivity to CLM, a specific DXS inhibitor, compared with that of the wild-type [5]. Under stress, especially heat-stress conditions, J20 promotes the degradation of DXS. On the other hand, cpHSP70 and reversible DXS can interact with the HSP100 chaperone ClpB3; the latter protein can synergistically contribute to refolding DXS back to its active form [30].

J20, J8 (AtDJC22, At1g80920) and J11 (DJC23, At4g36040) knock-out mutants were analyzed by Chen and coworkers [31]. These mutants all showed lower photosynthetic efficiency, the destabilization of photosystem II complexes, and unbalanced the redox reactions in chloroplasts. The *AtJ8* knock-out mutant has a lighter effect on photosynthetic parameters than the *AtJ11* or *AtJ20* knock-out mutant [31]. It was assumed that AtJ8, AtJ11 and AtJ20 possess at least partially redundant functions, but also specific functions, respectively. There are three J-proteins with small molecular masses in *Arabidopsis* chloroplast, which are AtJ8, AtJ11 and AtJ20, respectively. These three J-proteins can assist HSP70 chaperone proteins to ensure the activity of Rubisco (Ribulose bisphosphate carboxylase oxygenase) by correctly folding and assembling the enzyme [31,32]. When one of the three J-proteins is knocked out, the activity of Rubisco, which fixes carbon dioxide in photosynthesis, is negatively affected. Therefore, the ability of *atj11* or *atj20* single mutant to fix carbon dioxide will be greatly reduced with the decrease in enzyme activity. The carbon dioxide assimilation ability of the *atj8* mutant was slightly lower than that of the wild-type [31]. At the same time, the electron transfer pathways mediating ribulose-1,5-bisphosphate regeneration and trisaccharide phosphate metabolism in the *atj8* mutant are blocked, which limits the carbon reaction pathway of photosynthesis [31,33]. It was found that with the content of PSⅡ–LHCⅡ supercomplex, which decreased in a single mutant, the stability of PSⅡ dimer weakened and the number decreased significantly under high light. It can be seen that these three J-proteins maintain the efficiency of photosynthesis and stabilize the photosynthetic pigment-protein complex of the thylakoid membrane [31]. In addition, when any of the three chloroplasts-targeted J-proteins in *Arabidopsis* is knocked out, the redox reactions easily lose balance, since the dynamic regulation ability of redox reactions in chloroplasts is negatively affected, which increases the tolerance of *Arabidopsis* to oxidative stress caused by high light or methyl viologen [31,34,35].

## 5. Choloroplast J-Proteins in *Chlamydomonas reinhardtii*


There are five chloroplast DnaJ homologs (CDJ) proteins in *Chlamydomonas*, namely CDJ1 to 5. CDJ1 is a plastidic protein containing a zinc-finger domain, which localizes to the soluble matrix part, thylakoid and low-density membrane of chloroplast. High temperature only can weakly induce the expression of the *CDJ1* gene; therefore, the CDJ1 protein is only slightly increased under heat treatment. Solid experiment results showed that HSP90C and HSP70B form a complex in advance and then bind to CDJ1 [36]. The protein that interacts with both HSP70B and its cochaperone CDJ2 was identified by mass spectrometry as vesicle-induced protein (VIPP1) in plastids, which is essential for thylakoid biogenesis. Therefore, CDJ2 can specifically recognize and bind the substrate protein VIPP1 and recruit it to HSP70B, thus, participating in thylakoid membrane biogenesis [37]. CDJ3 and CDJ4 are weakly expressed and appear to be localized to the stroma and thylakoid membranes, respectively [38]. CDJ3 is strongly induced by light, and CDJ5 was also found to be light-inducible. The homologues of CDJ3-5 also can be found in green algae, moss and higher plants. CDJ3-5 all have special domains called bacterial-type ferredoxin domains. Since they all have redox-active Fe-S clusters, CDJ3-5 can activate ATPase activity on HSP70B through its J-domain and recruit HSP70B to participate in chloroplast Fe-S cluster biogenesis [39]. CDJ3-5 are most similar to the Fd domain-containing DJC76 clade, including DJC76, DJC77 and DJC82 of *Arabidopsis* [8]. Therefore, it is interesting to explore the function of DJC76 clade proteins, since DJA5 and DJA6 are involved in plastid Fe-S cluster biogenesis [14]. Determining the relationship between DJC76 clade proteins and DJA5/6 requires further analysis.

## 6. DJC31 and DJC62

The localization of DJC31 and DJC62 is relatively special, and whether they can be localized in chloroplasts is related to the integrity of their proteins. DJC31 and DJC62 are two structurally similar J-proteins that both carry clamp-type tetratricopeptide repeat domains (TPRs) and belong to the class C type of J-protein. When *DJC31* or *DJC62* in *Arabidopsis* was knocked out, the phenotype of the single mutant has little change compared with the wild-type. When both of them were knocked out, the morphology of roots, leaves, flowers and siliques were all abnormal, indicating that DJC31 and DJC62 are important for maintaining the morphology of plants [11]. In addition, *djc31djc62* double mutant is more drought tolerant than the wild-type, and hypersensitive to ABA. Previously, DJC31 and DJC62 were predicted to localize either to the nucleus or the chloroplast [40]. Further, chloroplast import experiments found that truncated forms of DJC31 and DJC62 could be imported, indicating that both of them are located in the chloroplast [8]. Surprisely, Dittmer and co-authors recently discovered that both DJC31 and DJC62 are located to the endoplasmic reticulum membrane, which was validated by detecting the two proteins in isolated chloroplasts and microsomal membranes [11]. The TPR domains of DJC31 and DJC62 share the conserved K5N9-N6-K2R6 motif with the human HSP70 and HSP90 co-chaperone TPR2 (also known as DNAJC7). This motif forms a carboxylate clamp that recognizes the EEVD motif in cytosolic HSP70 and HSP90 chaperones [40,41]. Indeed, *Arabidopsis* DJC31 and DJC62 might act as co-chaperones of HSP70-1 and HSP90-2 through interaction in the cytoplasm [11]. Although the evidence is solid for the localization of DJC31 and DJC62 shown by Dittmer and co-authors, it is worth mentioning that DJC62 possesses three different splice variants. It is possible that the shorter version of alternative splice may import into chloroplasts. Therefore, DJC31 and DJC62 are included in this review.

## 7. Plastid-Localized J-Like Proteins

*Orange* (*OR*) was cloned from orange cauliflower mutant, melon fruit and carrot roots, which is required for carotenoid accumulation [42,43,44]. OR is a J-like protein that lacks the J-domain and the C-terminal domain of classic J-protein, and contains a DNAJ-type zinc-finger domain [42,43]. Plastid-localized OR is able to regulate a major rate-limiting enzyme of carotenoid biosynthesis, therefore, promoting carotenoid biosynthesis [45]. In addition, it also regulates plastid preprotein import by interacting with Tic40 and Tic110, two key translocons for preprotein importing [13]. A gain-of-function mutation endow OR the function of promoting chromoplast biogenesis [46]. OR^his^ variant directly interacts with ACCUMULATION AND REPLICATION OF CHLOROPLASTS3 (ARC3), and the interaction interferes with the interaction of ARC3 and PARALOG of ARC6 (PARC6) [47]. Both ARC3 and PARC6 are crucial regulators of plastid division, and their interaction is important for plastid division. Therefore, OR^his^ can also regulate chromoplast number. The nucleus-localized OR interacts with the transcription factor TCP14 and represses its transactivation activity, therefore, repressing chloroplast biogenesis in the etiolated cotyledons of *Arabidopsis* [48]. Interestingly, OR is present in the nucleus only in etiolated tissue in darkness. When plants are exposed to light the protein relocates into fully developed chloroplasts [48].

ARC6 is another plastid-targeted J-like protein. ARC6 can promote chloroplast division in plant cells through the coordination of the filamenting temperature-sensitive Z (FtsZ) ring and ARC5 ring. When ARC6 is defective in *Arabidopsis thaliana*, the positions of plastid division proteins FtsZ1 and FtsZ2 change abnormally and a large number of short and disordered FtsZ filaments are formed in the chloroplast. The aberrant assembly of the FtsZ ring would result in aberrant plastid division such that the mesophyll cells would contain only one or two severely enlarged chloroplasts. It can be inferred that ARC6 is critical for the localization and assembly of FtsZ rings during plastid division [49]. In vascular plants, a specific ARC6 analogue is localized downstream of ARC6, called PARC6. PARC6 mediates the localization of PDV1 by interacting with it. PARC6 may restrain FtsZ assembly by interacting with ARC3, while ARC6 boosts FtsZ assembly. It was found that PARC6 and ARC6, as antagonistic regulators of FtsZ dynamics, play an essential role in chloroplast division [50].

Chaperone-LIKE PROTEIN OF POR1 (CPP1) interacts with protochlorophyllide oxidoreductase (POR) isoforms through its J-like domain on thylakoids to promote the stability of POR. The light-dependent POR can catalyze the reduction in protochlorophyllide to chlorophyllide and regulate the synthesis of chlorophyllide. Therefore, when CPP1 was deficient in *Arabidopsis thaliana*, the stability of the POR protein was weakened and the content of POR protein was decreased, which resulted in the inhibition of chlorophyll synthesis [51].

Thylakoids, the inner membrane system that contains the photosynthetic apparatus, are critical for chloroplast biogenesis. The signal recognition particle (SRP) pathway can target LHCB proteins and integrate them into the thylakoid membrane, mainly in rosette leaves, thus, promoting the formation of thylakoid. The *Snowy Cotyledon 2* (*SCO2*) encodes a J-like disulfide isomerase in chloroplasts. Since it does not interact with SRP54 and FtsY(a chloroplast homologue of the SRP receptor), which are proteins involved in the SRP pathway, SCO2 participates in thylakoid biogenesis through another pathway. It is known that SCO2 is integrated into the thylakoid membrane system by attracting and incorporating the chlorophyll *a*/*b* binding protein LHCB through its interaction with LHCB. When SCO2 is knocked out in *Arabidopsis thaliana*, the transport process of vesicles in the chloroplasts of cotyledons is disturbed, and the vesicles accumulate in the circular ends of chloroplasts, causing the internal disorder of thylakoids [52]. In addition, SCO2 has been shown to interact with some subunits of photosystem I (PSI) and PSII to participate in the assembly and repair of PSII [53].

Photosystem I Assembly 2 (PSA2) is found in the thylakoids of photosynthetic organisms such as green algae and plants. PSA2 interacts with PsaG through its DNAJ-type zinc-finger conserved domain to form a complex that mediates Thiol transactions in thylakoids, thereby promoting the synthesis of proteins and the assembly of proteins and pigments in PSI [54].

## 8. Discussion and Perspectives

In the chloroplast of *Chlamydomonas*, only a few J-proteins have been found [36,37,38,55], while it is reported that *Arabidopsis* contains at least 18 J-proteins and many J-like proteins in the chloroplasts [8,9,11,12]. Subsequent studies showed that DJC31 and DJC62 were not located in chloroplasts. Recently, we found that AtDJC78 could import into chloroplasts and interact with cpHSP70-1 through its C-terminal [12]. Therefore, we tend to believe that there are 18 J- proteins in *Arabidopsis* chloroplasts. Canonical J-proteins have high conserved J-domains, indicating their evolutionary conservation. Moreover, the single J-protein deficiency often has no obvious phenotype, which also raises difficulties and challenges when analyzing their function. According to the research progress summarized in this paper, DJA5 or DJA6 single mutant does not have a distinct phenotype, while the *dja6/dja5* double mutant leads to a lethal phenotype. Similarly, the double knockout of DJC31 and DJC62 caused severe defects in growth and development, while *djc31* or *djc62* did not show an obvious difference with that of the wild-type [11]. These findings indicate that functional redundancy may be a feature of some chloroplast J-proteins. Further studies are required to support this idea. On the other hand, J-proteins were constantly differentiated during the process of evolution too. It also suggested that J-proteins were endowed with new specific functions. As we mentioned above, DJC75 was involved in seed germination in darkness, DJC22 played roles in photosynthesis, and DJA5 and DJA6 function as the iron donor for Fe-S cluster biogenesis in the plastid. It is possible that J-proteins are involved in more undemonstrated bioprocess of higher plants.

DJA6 and DJA5 were found to play a key role in iron utilization during Fe-S cluster biogenesis in the chloroplast [14]; moreover, it cannot be ruled out whether DJA6 and DJA5 have other functions. For example, as conserved J-proteins in cyanobacterium, glaucophyte, green algae and higher plants, DJA5 and DJA6 remain in the role of co-chaperones of cpHSP70s that function in plastidial protein quality control. If so, their substrates need a further demonstration. DJA4 and DJA7 displayed a similar structural arrangement of their domains with DJA5 and DJA6; however, the functions of DJA4 and DJA7 remain unknown. Moreover, in *Chlamydomonas*, the chloroplast DNAJ-like proteins CDJ3–5 also have redox-active Fe–S clusters [38]. This suggests that there may be an evolutionary relationship between DJA5/6 and CDJ3–5. Recently, it was found that DJC75 and DNAJD15 cooperated with each other and participated in nitrate-promoted seed germination in the dark [26,27]. These findings provide a clear example that J-like protein can cooperate with J-proteins to synergistically regulate metabolic activity.

The network for protein quality control in plastid is important for the repair or degradation of inactivated or aggregated proteins caused by environmental stress. As the co-chaperones of cpHSP70s, different J-proteins can identify the specific substrates, and bring the substrates to HSP70. At the same time, they are also able to improve the binding activity of HSP70s to substrate proteins. It is crucial to demonstrate each substrate of every J-protein; therefore, it becomes possible to draw the network of plastid protein quality control.

## Data Availability

Not applicable.

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
