# Peer review of "Research Progress in J-Proteins in the Chloroplast"

_genes, 2022, doi:10.3390/genes13081469_

Round 1

Reviewer 1 Report

Chloroplast J-proteins and their related J-like-proteins are essential components of the proteostasis machinery present in plastids. There are recent works that describe specific functions for these family of chaperones and co-chaperones in chloroplast, regulating for instance photosynthesis, metabolism, stress resistance, and chloroplast development. The manuscript by Zhao and co-authors review the current knowledge about these interesting players of diversified functions. The manuscript provides a useful overview of some J-proteins. However, the text could be improved in terms of nomenclature and should focus on the proven cpDNAJs. Besides, it would be interesting to cover also DNAJ-like proteins family (actually counting with more publications that the canonical ones) as well as chloroplast J-proteins in the algae Chlamydomonas.

Major comments:

11.  The nomenclature of this family of proteins is somehow controversial but the most accepted names are DNAJ proteins, J-proteins and J-domain proteins (Kampinga & Craig 2010 Nature Reviews; Kampinga et al. 2019 Cell Stress Chaperones). This reviewer thus suggests to introduce a hyphen and call them “J-proteins” instead of “J proteins” in this manuscript. In the abstract “DnaJ” should also be replaced by “DNAJ” since the former is usually reserved to the E. coli protein after which the family was named. Besides, in line 25 it would be better to indicate that “DNAJ proteins (also called HSP40) and not the other way around. The term Hsp40 should be avoided in line 40 since actually they have been called in this manner exclusively in yeast literature. Other nomenclature that could be improved are “chloroplast protein quality control (cpPQC)” instead “CPQC” in line 36, “chloroplast heat shock proteins (cpHSP)” instead “cpHscs”, and “chloroplast heat shock protein 70 (cpHSP70) instead “cpHsc70.1 in line 38 and thereafter (similar for cpHSP70.2). HSC70 is usually exclusively used for cytosolic HSP70 in plants.

22.  It is difficult to understand why DJC31 and DJC62 are included in this review. It is true that initially they were described as one of the 19 putative chloroplast J-proteins in Arabidopsis (Chiu et al. 2013 PLOS One). In this work the authors isolated intact chloroplast and performed chloroplast import experiments with in vitro translated proteins. However, for DJC31 and DJC62 they could not show successful import of the full-length proteins into the chloroplast. Exclusively C-terminally truncated forms appeared to be imported into chloroplast. Later works by Dittmer et al 2021 (Journal of Cell Science) actually demonstrated that both proteins localized at the cytosolic side of the endoplasmatic reticulum membrane. Moreover, they are able to interact in the cytosol by BiFC with cytosolic forms of HSP70 and HSP90. In fact, DJC31 and DJC62 are very close in phylogenetic analysis to cytosolic TPR2/DNAJC7 (Brychzy et al. 2003 EMBOJ). Thus, there J-proteins are clearly not plastidial components of the proteostasis machinery.

33.  In connexion with the previous comment, references to the work by Prof. Schroda’s group with chloroplast J-proteins in chloroplasts (CDJ1, CDJ2, CDJ3, CDJ4, and CDJ5) is missing in this manuscript. Specially interesting could be the work by Dorn et al. 2010 BiochemJ, in wich they described how CDJ3 and CDJ4 might be involved in Fe-S cluster assembly. The J-protein most similar to these isoforms in Arabidopsis is DJC77 (Chiu et al. 2013 PLOS One). Interestingly, DJA5 and DJA6 have been found to participate in Fe-S cluster biogenesis (Zhang et al. 2021 EMBOJ) as well discussed by in this text. Thus, it would be interesting to discuss about putative redundancy/specificity of these components.

44.  There are more than 50 DNAJ-like/DNAJ-related proteins encoded in the genome of Arabidopsis (Pulido & Leister 2017 New Phytologist). While canonical J-proteins are classified into A-C types, non-canonical J proteins have been classified into D-F types, according to the domain conservation. It is surprising then that the authors have decided to review the current knowledge only of ORANGE/DNAJE1.4 and no other chloroplast components of the family. Other examples of characterized DNAJ-like proteins are ARC6/DNAJD9 and PARC6/DNAJD10, assembly factors in the formation of the FtsZ filaments in the chloroplasts and essential for chloroplast division (Vitha et al. 203; Glyn et al. 2009); CPP1/DNAJD11, a chaperone of POR1 regulating chlorophyllide biosynthesis; SCO2/DNAJE1.5, assembly factor of photosystem II (Tanz et. Al. 2012 Plant Journal; Zagari et al. 2017 Mol Plant); PSA2, assembly factor of photosystem I (Fristedt et al. 2014 JBC). There are more examples of chloroplast DNAJ-like proteins, it would be worth extending this review to these DNAJD-F families (partially reviewed by Pulido & Lesiter 2017 New Phytol and Tamadaddi et al. 2022 Plant Cell Reports), clearly explaining that they have assembly factor activity even though they do not interact with HSP70.

Specific comments:

11.  “More than 10 J proteins” in the abstract (line 11) sounds a little bit vague, specially when latter it is stated that there are at least 19 (line 44) in reference to Chiu et al. work. As previously discussed, DJC31 and DJC62 should be excluded from this list. Please rephrase to “around 17 J-proteins” or equivalent.

22.  In lines 47-48 describing the three (A-C) types of canonical J-proteins it would be better to quote a general review (such as Kampinga et al. Cell Stress Chaperones 2019) instead of Chiu et al. PLOS One that it is focused on chloroplast J-proteins.

33.  As previously mentioned, many J-like proteins have been actually characterized. Please correct lines 55-56.

44.  In the paragraph related to DJC75 and DNAJD15 (lines 114-118) it would be interesting to highlight that this is a clear example of collaboration between a J-protein and a J-like-protein, even though only DJC75 might be able to interact with cpHSP70.

55.  The sentence “OR is located to both nucleus and plastids [41]” (line 174) may be a little bit misleading. It seems that the protein is dual-localized in both nucleus and plastids at the same time. However, OR is present in the nucleus only in etiolated tissue, in darkness. When plants are exposed to light the protein relocates into fully developed chloroplasts. For this reason, maybe it would be better to relocate this sentence at the end of the paragraph when this work is cited again (lines 182-184), explaining in detail this alternative nucleus/chloroplast localization of the DNAJ-like protein.

66.  It is not correct that OR-his interfere with the interaction between ARC3 and ARC6/PARC6 (lines 179.181). Please correct this paragraph.

77.  Please quote Chlamydomonas work in line 186.

88.  Please quote Pulido & Leister 2017 New Phytol besides “[9]” in lines 187-188 to cover the DNAJ-like proteins.

99.  They J domain is highly conserved in canonical DNAJ proteins (types A-C) in all J-proteins independently if they are located in chloroplasts or other cellular compartments. Please rephrase lines 188-189.

110.         There are just a few examples of chloroplast canonical DJA5&DJA6 and non-canonical ARC6/DNAJD9 & PARC6/DNAJD10 and ORANGE/DNAJE1.4 & ORANGE-like/DNAJE1.8, that have been demonstrated to have a redundant role. However, the large majority of chloroplast DNAJ and DNAJ-like proteins may be encoded by only one gen in Arabidopsis. Thus, it is good to mention that some might be redundant but it would be better to avoid the idea that “redundancy may be a general feature of chloroplast J prtoeins (lines 195-197).

Author Response

Dear reviewer,

We would like to thank you for providing helpful suggestions and comments. We have responded to your comments as written below.

We hope that you find our revision satisfactory.

Reviewer 2 Report

  • This paper deals with J Proteins in the Chloroplast, which are widely distributed in major cellular compartments. They summarize recent research advances on the roles of J proteins in the chloroplast. Overall, the authors have made a good attempt at reviewing J proteins. 

Author Response

Dear reviewer,

Thank you for your reading and comments, and we will continue to revise and improve the manuscript.

Reviewer 3 Report

Research Progress in J Proteins in the Chloroplast is an review article that consists of seven chapters: the introduction, five chapters that review the available literature, and last, which is discussion and perspectives. The article is quite concise and cites a small number (44) of literature. Not enough for review in my opinion. Therefore, the manuscript require a major revision.

Chapter 3 ‘DJC31 and DJC62 assist plants in dealing with abiotic stress’ reviews only two proteins and could be further elaborated. I am missing information about thermotolerance and connections with ATPase activity. There are more available literature considering J proteins and their role in dealing with abiotic stress.

Also chapter 6 ‘Plastid-localized OR regulates carotenoid biosynthesis, plastid preprotein import and chromoplast biogenesisis’ is a bit brief. There are more research about other DnaJ like zinc finger proteins in chloroplasts.

My suggestion for Author’s consideration would be the change of chapter titles. In my opinion they should be focused on the phenomenon. While their content should include an overview of the entire known, or at least the most important literature.

Minor suggestions:

The article has a clear structure, but needs to be checked for typos and stylistic errors. For example:

·         line 37 'to degradate or reactive the damaged or misfolded proteins', I guess authors meant ‘degrade’ and 'reactivate'

·         lines 66-67 'functions 66 of DJA5 and DJA6 are overlap' – should be ‘are overlapping’ or ‘overlap’ without auxiliary verb

·         line 190 ‘analyized’à analyze

·         line 206 ‘alage’à ‘algae’

 Additionally, sentence in lines 127-130 needs a citation.

Overall, this article is interesting and covers an important topic of chaperone proteins. In the age of climate change and constant rise in temperature it is essential to organize the knowledge of proteins that may influence the adaptation of plants to these changes. I would consider publishing after major revisions.

Author Response

(The authors gave the same response as above.)

Round 2

Reviewer 1 Report

The authors have properly addressed all the comments. The manuscript constitues an interesting overview on the field of chloroplast J-proteins and it may be ready for publication. Thank you.

Reviewer 3 Report

Thank you for taking into account my comments.